# Assessing Current Seismic Hazards in Irpinia Forty Years after the 1980 Earthquake: Merging Historical Seismicity and Satellite Data about Recent Ground Movements

**Aldo Piombino** [1,*], **Filippo Bernardini** [2] and **Gregorio Farolfi** [1]

1   Department of Earth Sciences, University of Florence, 50125 Florence, Italy; gregorio.farolfi@unifi.it
2   Istituto Nazionale di Geofisica e Vulcanologia (INGV), Sezione di Bologna, 40128 Bologna, Italy;
    filippo.bernardini@ingv.it
*   Correspondence: aldo.piombino@unifi.it

**Abstract:** Recently, a new strain rate map of Italy and the surrounding areas has been obtained by processing data acquired by the persistent scatterers (PS) of the synthetic aperture radar interferometry (InSAR) satellites—ERS and ENVISAT—between 1990 and 2012. This map clearly shows that there is a link between the strain rate and all the shallow earthquakes (less than 15 km deep) that occurred from 1990 to today, with their epicenters being placed only in high strain rate areas (e.g., Emilia plain, NW Tuscany, Central Apennines). However, the map also presents various regions with high strain rates but in which no damaging earthquakes have occurred since 1990. One of these regions is the Apennine sector, formed by Sannio and Irpinia. This area represents one of the most important seismic districts with a well-known and recorded seismicity from Roman times up to the present day. In our study, we merged historical records with new satellite techniques that allow for the precise determination of ground movements, and then derived physical dimensions, such as strain rate. In this way, we verified that in Irpinia, the occurrence of new strong shocks—forty years after one of the strongest known seismic events in the district that occurred on the 23 November 1980, measuring Mw 6.8—is still a realistic possibility. The reason for this is that, from 1990, only areas characterized by high strain rates have hosted significant earthquakes. This picture has been also confirmed by analyzing the historical catalog of events with seismic completeness for magnitude M ≥ 6 over the last four centuries. It is easy to see that strong seismic events with magnitude M ≥ 6 generally occurred at a relatively short time distance between one another, with a period of 200 years without strong earthquakes between the years 1732 and 1930. This aspect must be considered as very important from various points of view, particularly for civil protection plans, as well as civil engineering and urban planning development.

**Keywords:** Irpinia; seismic hazard; earthquake; strain rate; GNSS; InSAR

## 1. Introduction

This study is based on the analysis of a fine-scale ground velocity map of Italy determined by the fusion of Global Navigation Satellite Systems (GNSS) with synthetic aperture radar interferometry (InSAR) data derived from satellites [1]. The dataset derives from a period of observation between 1990 and 2012. The InSAR dataset is part of the "Piano Straordinario di Telerilevamento" (Special program for Remote Sensing, promoted by the Italian Ministry of Environment). Due to the quasi-polar orbit of the satellites, space-borne InSAR observations can only determine the East–West (E–W) and Up–Down (U–D) components of the movement of persistent scatterers. However, there are millions of scatterers that are unreachable, due to the fact that only a few hundred GNSS stations exist. The North–South (N–S) component is provided by a $C^2$ continuous bi-cubic interpolation function that is well suited to interpolate sparse GNSS stations displaced inside

the study site and surrounding areas. To do this it uses a hierarchical structure at different refinement levels.

The fusion of GNSS with InSAR is a method based on the calibration of InSAR with GNSS measures derived from permanent stations and survey campaigns [2,3]. The results are a coherent fine-scale ground velocity map with a spatial resolution that is unreachable using the previous velocity field maps determined with the GNSS technique alone. By using this technique, Farolfi, Piombino, and Catani [1] provided new information about the complex geodynamics of the Italian peninsula and thanks to the high spatial resolution of the ground movements map, identified interesting patterns of small areas with respect to the surrounding ones. Moreover, their work confirmed the division of peninsular Italy into two sectors, with opposed E–W components of movement in the Stable Europe Frame (Figure 1). This has been depicted by older studies based only on GNSS station movements ([4] and references therein): the western block (Tyrrhenian) is moving westward, while the eastern one (Adriatic) shows an eastward movement. The relative motion of these blocks implies their divergence; the effect of which is represented by the numerous currently active normal fault systems along the central and southern Apennines—which are close to the border between these two sectors—and the associated seismicity.

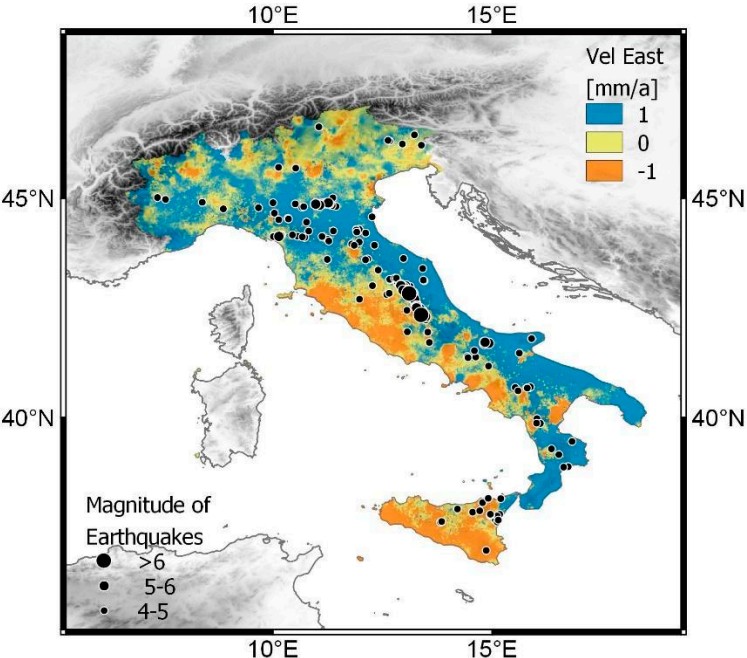

**Figure 1.** Map of the East component of the ground velocity field of the Italian Peninsula, derived from Global Navigation Satellite Systems (GNSS) and synthetic aperture radar interferometry (InSAR) during more than two decades of observation (1990–2017). The area of the Central Apennines presents major earthquakes from 1990 to present day. From the figure above, it is clear that the main seismogenic areas are linked to the boundary that divides the two blocks with opposite E–W components of velocity.

The Apennine chain, an approximately linear belt hosting the most rapidly slipping normal faults, and the most damaging earthquakes, are coincident with the areas in which the morphological surface height, when averaged on a horizontal scale of tens of kilometers, is greatest [5]. In this area, the first studies based on the relative movements of the GNSS stations have already determined a medium value of a ca. 3 mm/a extension, linked to the differential movements between the two blocks. This also allows the emplacement of melt intrusions along deep-rooted faults [6]—the last occurrence of this kind probably triggered the 2013/2014 Matese seismic swarm [7]—and the widespread emission of deep-originated $CO_2$ [8]. This regime is dissecting the former Cenozoic east-verging thrust belt related

to the west-dipping subduction of the Apulian lithosphere [9]. This compressive regime ended at 650 ka in the middle Pleistocene [10].

The E–W component of InSAR movements [1] has also confirmed the frame depicted by [11], in which the Ortona–Roccamonfina is not a single lineament, but a 30 km wide deformation channel: this channel is characterized by prevalent west-directed velocities in the stable Europe frame, nested in the Adriatic eastward-moving block.

The vertical component of the InSAR data highlights the current general uplift occurring in most of Southern Italy, even if this uplift is lower than in the Central Apennines (especially in the "Abruzzo Dome" [1]), confirming a wealth of the geological literature. Conversely, few areas show subsidence, mainly because of human groundwater exploitation. In this frame, the highest uplift values of the whole Southern Apennines—exceeding 1.8 mm/a—are present in the chain segment between Benevento and Potenza. This area of higher-than-surroundings uplift roughly corresponds to the Irpinia sector, in a belt just west of the Campania–Puglia border. Thus, it is possible to call this area the "Irpinian Dome" (Figure 2). The Ufita and Marzano faults represent the surface traces of the two different patterns of the East–West ground velocity component (Figure 3 (top)).

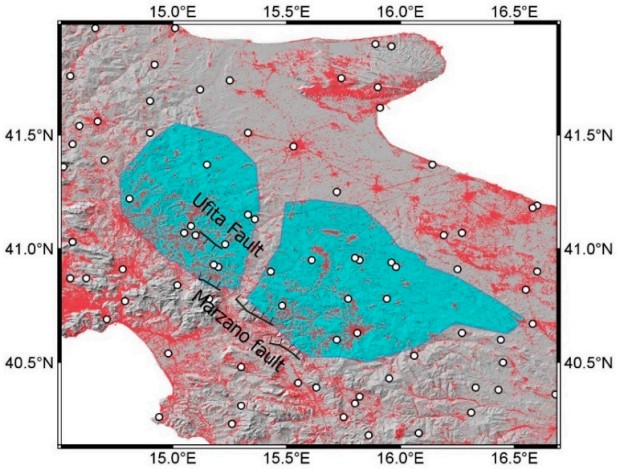

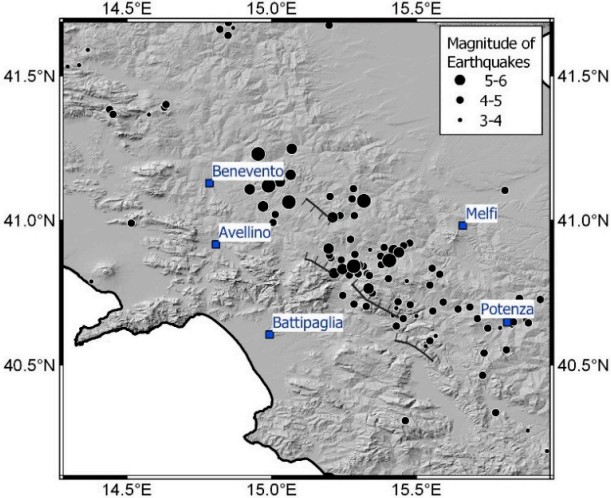

**Figure 2.** (**Top**) Map of the distribution of persistent scatterers (PS) (red points) and the GNSS permanent stations (white points) involved in the detection of ground surface movements of the study area. The main geodynamic features are represented in the background of the map: the Irpinian Dome is the cyan area and the Ufita and Marzano faults are represented with black hashed lines. (**Bottom**) Map of the main seismic events (black circles) that occurred from 1466 to 2017 with the main towns highlighted (dark blue squares).

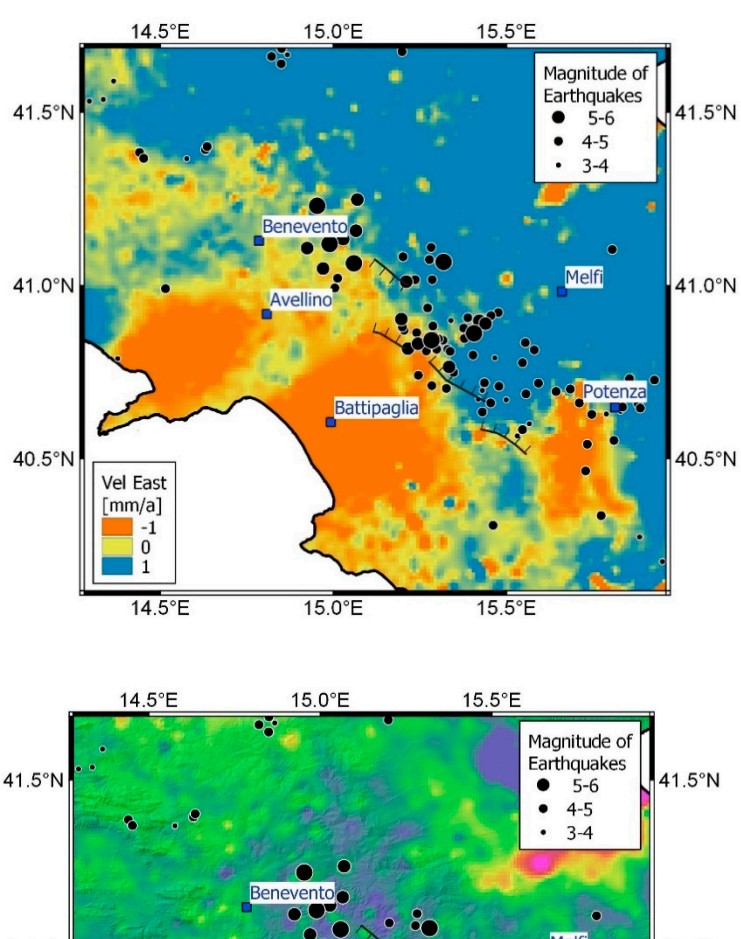

**Figure 3.** (**Top**) The East component of the ground velocity field of the Irpinia–Sannio area with the main earthquakes of M ≥ 3 occurred since 1900. For the earthquakes of M ≥ 4, the label represents the year of the occurrence. (**Bottom**) Map of the vertical component of the ground velocity in Irpinia with the main earthquakes that occurred from 1466 to 2017. The main towns are drawn and labeled with dark blue squares, and the main faults (see Figure 2 (top)) are represented with black hashed lines.

The uplifting area is divided into two different parts and, between them, exists a narrow corridor of lower uplift <1 mm/a, (Figure 3 (bottom)). It is interesting to note that this corridor is placed near the epicentral areas of the 1930 and 1980 earthquakes.

The geographical axis of the Irpinian dome is placed east of the main NE-dipping faults, on the surface projection of the hanging wall. Any useful information of the N–S component of the ground movement can be detected by InSAR satellites because their quasi-polar orbits only make the detection of vertical and E–W velocity components possible.

## 2. Relationship between Strain Rate and Earthquakes

The strain rate provides a measure of the superficial deformation, and for this reason, is useful information for studying and analyzing geodynamics. Many authors have produced strain rate maps of the Italian territory using GPS station data. In the last decade, for example, Riguzzi et al., (2012) [12] estimated the strain rate, using the GPS velocity solution, of the Italian area—provided by Devoti et al., (2011) [13].

Palano (2015) [14] carried out an analysis of the stress and strain-rate fields of Italy. He performed a comparison of GPS inferred strain-rate data and 308 stress datasets interpolated at each node of a regular grid.

Montone and Mariucci (2016) [15] provided an updated present day stress map for the Italian territory combining seismicity, data retrieved from a breakout analysis in deep wells, and fault data. Starting from this base Mastrolembo and Caporali (2017) [16] presented a direct comparison of the principal horizontal directions of stress and strain-rate directions of extension, estimated at the position of each stress measurement in their data set. For this, they used GPS data coming from over 500 stations distributed on the Italian peninsula, however, they did not provide a general map.

This work instead benefits from a new fine-scale strain rate field of the whole continental Italy and Sicily (Figure 4) [17], determined from the surface ground movements map obtained by the satellite InSAR observations between 1990 and 2012 [1]. The two-dimensional velocity gradient tensor is calculated by applying the infinitesimal strain approach [18,19] with a grid of 20 km × 20 km. The known horizontal incremental velocity vector $V_i$ of the *i*-vertex polygon is defined as:

$$V_i = A_i + \frac{\partial V_i}{\partial x_j} x_j = A_i + t_{ij} x_j \tag{1}$$

where $A_i$ is the unknown velocity at the origin of the coordinate system, $x_j$ is the position of the station, and $t_{ij}$ is the displacement gradient tensor. Following the tensor theory, we separated the second-rank tensor into a symmetric and an anti-symmetric tensor. Then, $t_{ij}$ can be additively decomposed as follows:

$$t_{ij} = \frac{(t_{ij} + t_{ji})}{2} + \frac{(t_{ij} - t_{ji})}{2} = e_{ij} + \omega_{ij} \tag{2}$$

The symmetric and anti-symmetric parts of the infinitesimal strain rates can be associated with the infinitesimal strain $e_{ij}$ and rotation $\omega_{ij}$ tensors. Principal strains $e_1, e_2$ were computed as:

$$e_1, e_2 = \frac{1}{2}(e_{ii} + e_{jj}) \pm \frac{1}{2}\sqrt{(e_{ii} - e_{jj})^2 + 4e_{ij}^2} \tag{3}$$

and the horizontal second invariant of the strain rate (*SR*) tensor was also evaluated as the scalars and is presented in Figure 4:

$$SR = \sqrt[2]{e_1^2 + e_2^2} \tag{4}$$

The determination of the second invariant of the strain rate provides important additional information to support the analysis of the geodynamics and the earthquake distribution of the study area. A recent study [17] based on the analysis of the seismic events that have occurred since 1990 in the Italian peninsula, shows that the probability of earthquakes occurring is linked to *SR* by a linear correlation. More specifically, the probability that a strong seismic event will occur doubles with the doubling of *SR*. Then, the *SR* is used as an independent and quantitative tool to spatially forecast seismicity.

The results of this study agree with these former studies, especially for the detection of the high strain rate along the Central and Southern Apennines axis and in Northern Sicily [19].

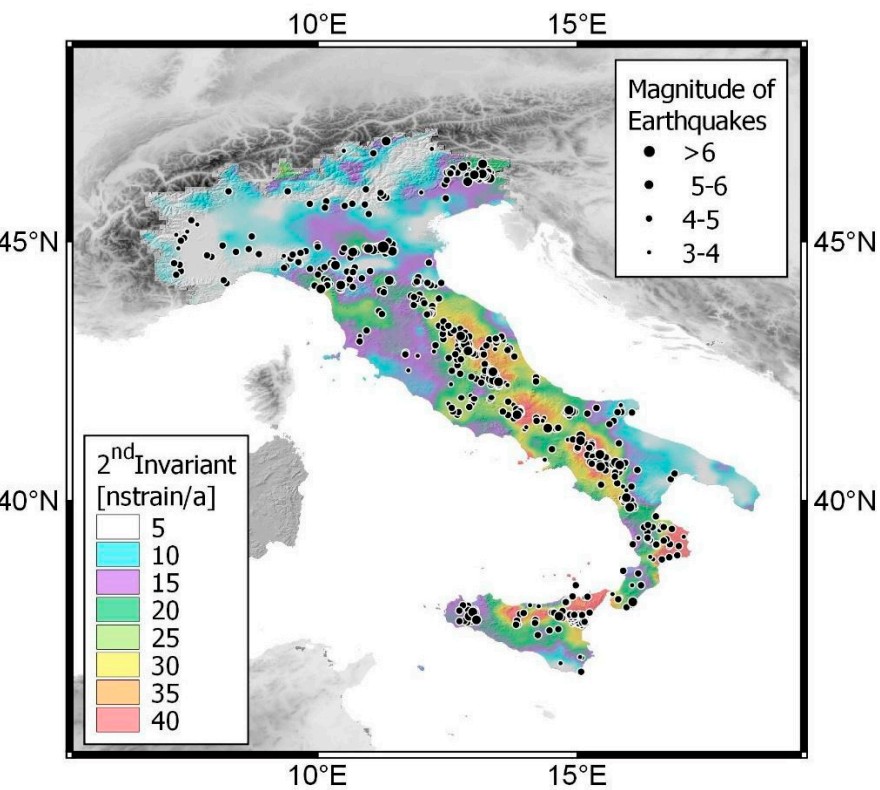

**Figure 4.** Map of the horizontal strain rate field of the Italian peninsula, determined by an infinitesimal approach from the horizontal velocity field derived from GNSS and InSAR during more than two decades of observation (1990–2017). Main earthquakes that occurred from 1990 to 2017 are represented on the map.

This new theory, based on observables, identifies significant earthquake (M > 5.5) prone areas with high strain rate areas. It gives a new perspective for the interpretation of recent earthquakes and this theory also predicts the events that occurred after the observation period of the study. For example, in the outer side of the Alps, the strongest earthquakes recorded in the past decades occurred in just two of the four areas of this sector showing high strain rate value: the M 5.3 20 December 1991 Graubunden [20] and the M 5.3 22 March 2020 Zagabria earthquake [21]. The others selected areas located around the cities of Marseille and Innsbruck. Furthermore, after 1990 in Italy, all earthquakes with M > 5.1 and a hypocenter less than 15 km deep only occurred in areas showing a high strain rate: 1997, 2009 and 2016 Central Apennines seismic sequences, the 2012 Emilia earthquakes, and the 2013 Lunigiana earthquake (on the surface of the Po Plain the *SR* is lower than in the buried and seismic Apenninic units because of its attenuation in the plastic Neogene sedimentary cover).

Outside these areas no shallow significant earthquakes occurred until 1990, even though strong events occurred after 1940, such as Friuli (the M 6.5 6 May 1976), Western Sicily (the Mw 6.4 15 January 1968), and Valais (the M 6.1 25 January 1946). In addition to these high strain rate areas that have been hit by strong earthquakes, there are others that, while showing high values of this value, have not been hit by relevant earthquakes since 1990. In recent years, only areas characterized by high strain rates have been affected by significant earthquakes, therefore, it is not unreasonable to empirically hypothesize that significant seismic events of the next decades have a greater chance of occurring only in the areas characterized by high strain rates. The year 1990 is taken as a milestone because, after beginning the survey in 1991, the former earthquakes do not influence the data.

### 3. Stain Rate in Irpinia

Irpinia is one of the main areas of the core of the Central and Southern Apennines chain. The differential movements between the two blocks, in which the Italian peninsula is divided imply a medium strain rate of 50 nstrain/a [5]. Here, the main fault systems are the Ufita, Monte Marzano, and Caggiano faults [22] (Figure 2 (top)). However, deformation is also linked to faults with a highly different orientation, well constrained in the historical record. For this issue, it is interesting to note that the focal mechanism solutions of the 1930 and 1962 earthquakes are significantly different from the kinematics of the typical large earthquakes that occurred along the crest of the Southern Apennines. Instead, these are well-fitted by the Mw 6.9 23 November 1980 earthquake, caused by predominant normal faulting along NW–SE-striking planes. The fault linked to the Mw 6.7 23 July 1930 earthquake is blind and its magnitude and focal mechanism are debated ([23] and references therein). Many focal mechanisms have been proposed, from a "classical" NW–SE to an ESE–WNW striking plane. These belong to an array of oblique dextral slips on the EW-trending planes crossing the whole Southern Apennines which is dissecting the orogen in various contiguous sectors. The level of the transcurrent component is debated as well. However, the effects of the earthquake presented in [24] fit better with a NW–SE striking fault.

The 1962 sequence is composed of three different shocks at 18:09, 18:19, and 18:44 UTC, the second being the most destructive (Io IX MCS, Mw 6.1, [25]). Additionally, identification of the faults responsible for these earthquakes is difficult because of the lack of reported surface faulting. Only in 2016 was a reliable focal mechanism produced [25] with two solutions: dominant strike-slip rupture along a north-dipping, E–W striking plane, or along a west-dipping, N–S striking plane. Its depth is still controversial, varying between 7 and 35 km. Therefore, the focal mechanism solutions of the 1962 earthquakes are significantly different from the kinematics of the typical large earthquakes occurring along the crest of the Southern Apennines, well-fitted instead by the Mw 6.9 23 November 1980 earthquake, caused by predominant normal faulting along NW–SE-striking planes.

Irpinia is one of the areas in Italy showing a higher strain rate (Figures 4 and 5) during the 1991–2011 InSAR survey: currently north of it, in the Sannio sector, the strain rate is at a low level with a value of 20 nstrain/a 10 km north of Benevento. However, to the SE of Benevento, the value increases to 35 nstrain/a in less than 30 km at Grottaminarda, reaching the highest levels (48 nstrain/a) 15 km south of the epicenter of the 23 November 1980 earthquake. Therefore, Irpinia is still currently one of the areas with a higher strain rate in Italy, with values always >32 nstrain/a, and showing a maxima over the hanging wall of the Monte Marzano fault system. The southward strain rate dramatically drops to 35 nstrain/a near Polla. However, while north of Irpinia along the chain axis the value drops rapidly under 30 nstrain/a, the southward values remain above this value for much longer, up to the Pollino line (the border between Central Apennines and Calabria–Peloritani arc). In the picture of the EW-trending lithospheric faults dissecting the Apenninic orogen, these sudden strain rate drops north and south of Irpinia can be related to different strain rate conditions occurring in the adjacent sectors.

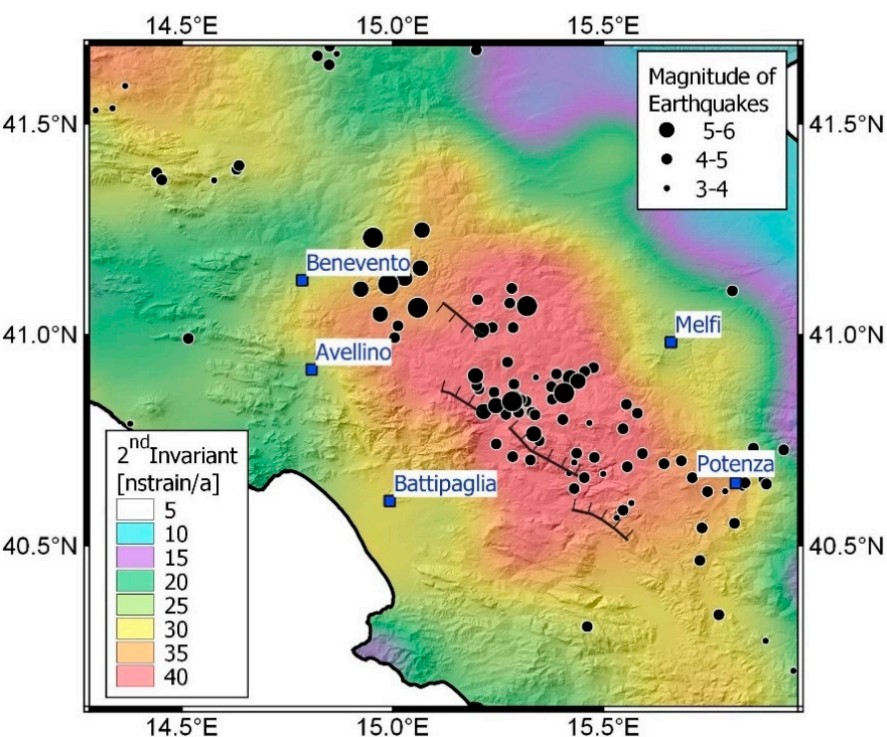

**Figure 5.** Map of the horizontal strain rate field of Irpinia derived from GNSS and InSAR during more than two decades of observation (1990–2017). The main earthquakes that occurred from 1466 to 2017 are represented on the map. The main towns of the area are drawn and labeled with dark blue squares, and the main faults (see Figure 2 (top)) are represented with black hashed lines. Since the strongest shallow events that occurred inland in Italy from 1990 to today are placed only in areas characterized by high strain rates [17], the high strain rate detected in Irpinia implies—from a theoretical point of view—a scenario where a new strong earthquake seems more likely. This can be somehow counterintuitive, because this area hosted most of the strongest earthquakes in southern Italy after 1908, in 1930, 1962, and 1980: only the Mw 6.4 1968 Belice and the Mw 6.0 1978 Patti gulf events (both in Sicily) reached similar magnitudes [26]. Only in the NE Sicily 1978 earthquake was the strain rate as high as in Irpinia. Therefore, from this point of view, we can hypothesize that in Irpinia, the probability of a new strong event is still very high.

## 4. The Historical Record of Earthquakes in Irpinia

Additionally, historical seismicity can allow this—somehow unexpected—statement, given the time intervals between Irpinian earthquakes. "Irpinia" is a historical–geographical area of southern Italy, located in the Campania region, approximately corresponding to the territory of the current province of Avellino, which in turn, largely recalls the historic province of Principato Ultra of the Kingdom of Naples.

The Irpinia area is one of the most seismically active sectors of the entire Italian territory. The seismogenic belt that runs along the Apennine chain, in fact, crosses the northern and eastern part of the province of Avellino, where strong earthquakes have frequently occurred over centuries.

The most important historical seismic events are placed in the hanging wall of the Monte Marzano fault system [22]. If we take a polygon with vertices at the coordinate points 41.314° N, 14.971° E; 41.105° N, 14.874° E; 40.739° N, 15.352° E; 41.056° N, 15.574° E (depicted in dark red in Figure 6 (bottom)), corresponding to the Apennine seismic belt site of the major historical and instrumental seismicity, the parametric catalog of Italian earthquakes CPTI15 [26] reports about twenty earthquakes with magnitude Mw ≥ 5.0, starting from the year 1000 (see Table 1). Of these, seven have a Mw between 6.0 and 6.8. It must be said that the catalog can be considered complete, for the strongest events (Mw ≥ 6.0) only for the last 400 years, namely from 1620 up to today [27]. From the

diagram in Figure 6 (top), it can be seen that until the end of the 17th century, the seismic history of the Irpinia sector is largely incomplete and poorly documented. This, obviously, is not because there were no earthquakes at all, but because only little and partial historical information about that area for those ancient periods exists today. Only a couple of earthquakes are known (in 1466 and 1517) to have occurred in this period, plus two events before the year 1000, which occurred in the year 989 and 62 CE [28]; thus, outside the reference window of the historical catalog. Both these events originated from the monte Marzano Fault [29]. The earthquake of 5 December 1456 [30] was deliberately not taken into consideration in the present study, because it is a complex event that affected a very large area of southern Italy, causing damage from Puglia to Abruzzo, and whose epicenter is not well located nor defined. Probably, that earthquake was made up of several shocks that occurred in different sectors of the central–southern Apennines a few days apart, and Irpinia was only one of the several areas that were struck [28].

**Table 1.** List of the main Irpinia earthquakes (Mw > 5.0) extracted from the CPTI15 catalog [26]. For the description of the various parameters see this catalog at https://emidius.mi.ingv.it/CPTI15-DBMI15/index_en.htm (accessed on 30 January 2021) As attested by Rovida et al. [27], this historical record can be considered complete since 1620 for M 6.0+ earthquakes.

| Year Mo Da | Epicentral Area | Lat | Lon | Io (MCS) | Mw |
|---|---|---|---|---|---|
| 1466 01 15 | Irpinia–Basilicata | 40.765 | 15.334 | 8–9 | 6.0 |
| 1517 03 29 | Irpinia | 41.011 | 15.210 | 7–8 | 5.3 |
| 1692 03 04 | Irpinia | 40.903 | 15.196 | 8 | 5.9 |
| 1694 09 08 | Irpinia–Basilicata | 40.862 | 15.406 | 10 | 6.7 |
| 1702 03 14 | Sannio–Irpinia | 41.120 | 14.989 | 10 | 6.6 |
| 1732 11 29 | Irpinia | 41.064 | 15.059 | 10–11 | 6.8 |
| 1741 08 06 | Irpinia | 41.049 | 14.970 | 7–8 | 5.4 |
| 1794 06 12 | Irpinia | 41.108 | 14.924 | 7 | 5.3 |
| 1853 04 09 | Irpinia | 40.818 | 15.215 | 8 | 5.6 |
| 1905 11 26 | Irpinia | 41.134 | 15.028 | 7–8 | 5.2 |
| 1910 06 07 | Irpinia–Basilicata | 40.898 | 15.421 | 8 | 5.8 |
| 1930 07 23 | Irpinia | 41.068 | 15.318 | 10 | 6.7 |
| 1962 08 21 | Irpinia | 41.248 | 15.069 | | 5.7 |
| 1962 08 21 | Irpinia | 41.230 | 14.953 | 9 | 6.2 |
| 1962 08 21 | Irpinia | 41.158 | 15.065 | | 5.3 |
| 1980 11 23 | Irpinia–Basilicata | 40.842 | 15.283 | 10 | 6.8 |
| 1980 11 24 | Irpinia–Basilicata | 40.811 | 15.268 | | 5.0 |
| 1981 01 16 | Irpinia–Basilicata | 40.890 | 15.439 | | 5.2 |
| 1982 08 15 | Irpinia | 40.832 | 15.244 | 6 | 5.3 |

A lack of seismic events in the historical record for a given area can be due to the following reasons:

(a)  an area of genuinely low long-term seismicity;
(b)  either the incompleteness or a too-short time-span of the earthquake catalog;
(c)  a quiescent period in an area characterized by temporal clustering, followed by a long recurrence interval [31].

The seismic history of Irpinia is better documented, starting from the end of 1600, and as minor events (4.0 ≤ Mw < 5.0) can be considered well documented only starting from the end of the 19th century (Figure 6)

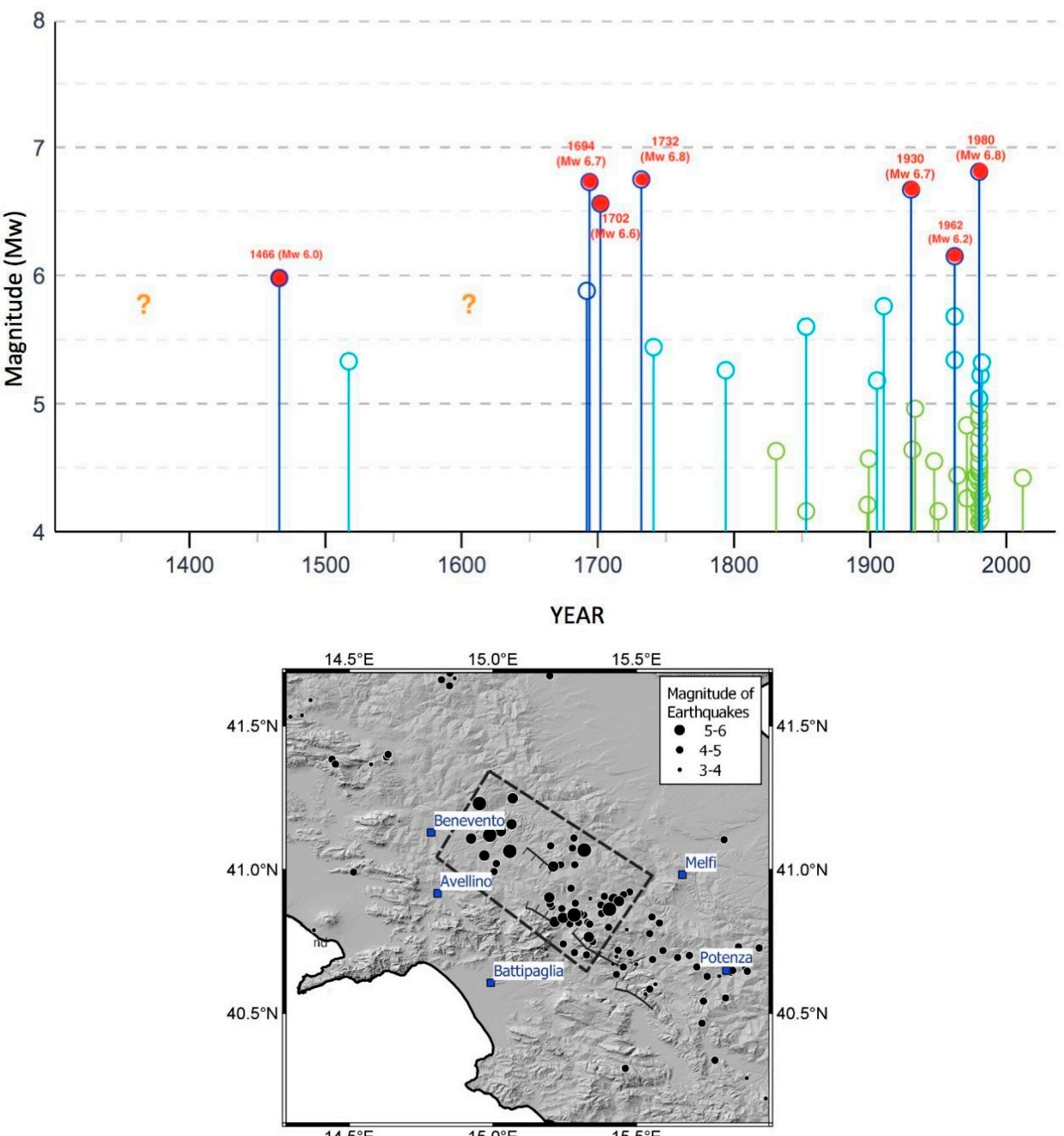

**Figure 6.** (**Top**) Seismic history of the Irpinia area as from the CPTI15 catalog [26]. The dark red polygonal area shows the Irpinian main seismic belt, the coordinates of its vertices are: 41.314° N, 14.971° E; 41.105° N, 14.874° E; 40.739°, 15.352° E; 41.056° N, 15.574° E. Such a historical record is well documented only for the last 400 years (namely since 1620 [27]), whereas for the previous centuries it is poorly known, with long timespans lacking information. The figure also shows that inside the catalog completeness span-time of 400 years (since 1620), there are a couple of evident clusters of strong earthquakes (Mw ≥ 6.0) which are 200-years apart. The first, between 1694 and 1732, when three M 6.5+ events occurred over a period of 38 years. The second, between 1930 and 1980, when three M 6.0+ events occurred over a period of 50 years. (**Bottom**). Map of the main historical seismic events of Irpinia from 1466 until 1982, reported in Table 1. The main towns of the area are drawn and labeled with dark blue squares, and the main faults (see Figure 2 (top)) are represented with black hashed lines.

From its seismic history it can also be seen that, over the 400 year time-span of seismic catalog completeness for M ≥ 6.0 events, in Irpinia, the strongest earthquakes (Mw ≥ 6.0) tend to group over time, spaced from long phases characterized by lower and less frequent

seismicity (Figure 6 (top and bottom)). At the turn of the seventeenth and eighteenth centuries, over a period of 40 years, Irpinia was affected by four damaging earthquakes, three of which occurred in just 10 years (1692, 1694, 1702, and 1732). Of these, the ones that occurred in 1694 (considered as a sort of twin of the 1980 earthquake), in 1702 and in 1732 were large events of Mw > 6.5. Each of these caused extensive destruction over large areas and many casualties. Another cluster of strong earthquakes is the one that hit the sector in the twentieth century, between 1930 and 1980 (three events with Mw ≥ 6.0 over a period of 50 years). So, Irpinia belongs to the belt of very high seismic hazard running along the Central and Southern Apennines (Figure 7).

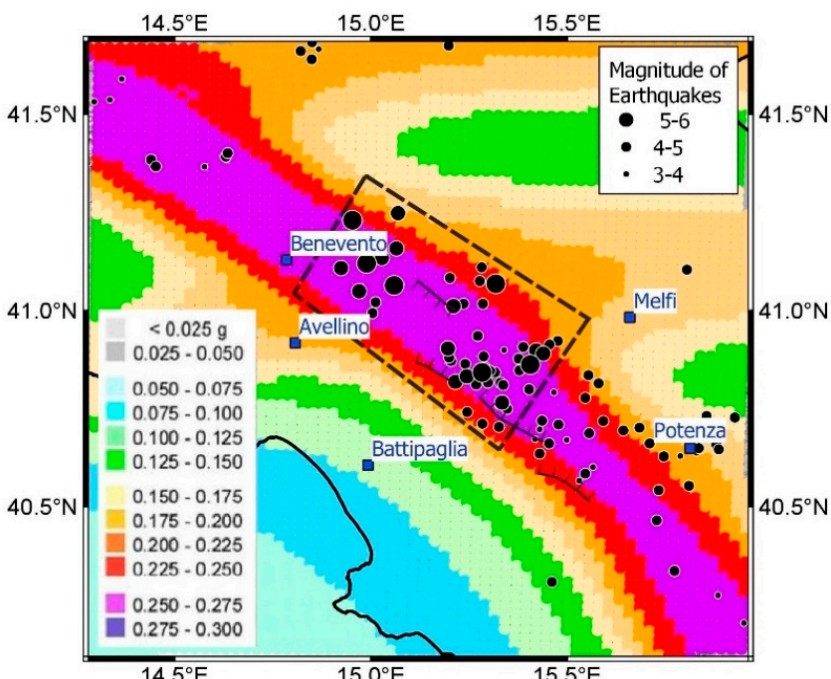

**Figure 7.** Map of seismic hazards in Irpinia (see [32]) and adjoining areas (colors in the background), derived mostly from the historical seismic records, as shown by the overlapping of strong seismic events within the map itself. The main towns of the area are drawn and labeled with dark blue squares, and the main faults (see Figure 2 (top)) are represented with black hashed lines.

It is unlikely that in the 200-year time-span between 1732 and 1930 there were large (M 6.0+) earthquakes in the Irpinia area, since these are not present in the historical record. In the same time interval, not just "minor" earthquakes are well documented in the very same area (i.e., the 1741 Mw 5.4, 1794 Mw 5.3, and 1853 Mw 5.6 Irpinian events; see Table 1, Figure 5), but strong events are also well known to have struck other adjacent Apennine areas (the 1805 Mw 6.7 Matese earthquake; those of 1851 Mw 6.5 and 1857 Mw 7.1 in Basilicata [26]). Therefore, it can be assumed that the historical seismicity of Irpinia has been characterized by periods of intense activity, with strong earthquakes over a few years or decades, interspersed with long periods of minor-to-moderate activity, with earthquakes of magnitude lower than 6.0.

The spatio-temporal clustering of earthquakes in the Southern Apennines is well documented in the scientific literature. By comparing the number of earthquakes on record in the last five to seven centuries, with the number implied by slip-rates on active normal faults averaged over 18 kyrs in the Southern Apennines, Papanikolaou and Roberts [33] demonstrated that the long history of earthquakes in the Italian Apennines may indeed contain evidence for earthquake clustering. In particular, according to Papanikolaou and Roberts [33], Irpinia and northern Basilicata show a very high number of earthquakes and this indicates that this area may be in a temporal earthquake cluster phase. Meanwhile,

the sector located slightly further south, up to the Pollino massif, could be in a temporary anti-clustering process.

The strain rate map in Farolfi et al. [17], in which the Irpinia–Basilicata sector is characterized by a much higher strain rate than the Pollino sector (and the intermediate sector, Vallo di Diano, shows intermediate values), fits well with these results.

In the Central and Northern Apennines, earthquake clustering is known to exist. For example, Tondi and Cello [34] observed a time interval of ca. 350 years among the beginning of seismic clusters in the Central Apennines Fault System. The current sequence, which started in 1997 and continued with the events of 2009 and 2016, has arrived on time if we consider that two main seismic clusters in the past began in the years 1349 and 1688 [35]. In the Northern Apennines, a major seismic crisis occurred between 1915 and 1921 [36], while in this area, the historical record before 1915 is composed of a few destructive events [26]. Additionally, the two-year period 2012–2013 showed a high level of activity, not only in the area of the Emilia seismic sequence, but also in the Garfagnana sector, accompanied by a high strain rate.

Currently, the most likely explanation for seismic clustering is the "stress transfer" between faults [36] and references therein, due to coseismic movement rearranging the Coulomb failure stress on other nearby faults [37]. However, this explanation falls short when there is the occurrence of an isolated, single event (such as the Mw 6.1 6 November 1599 Valnerina, and the Mw 6.4 13 January 1832 Valle Umbra earthquakes) that do not trigger a level of Coulomb stress transfer, resulting in strong earthquakes on other neighboring faults.

For other researchers, there is a sort of "domino effect" between the crustal blocks that make up the Apennines [38].

In conclusion, we suppose that the deformation rate value, as described in Farolfi et al. [18], represents a *conditio sine qua non* for the occurrence of strong earthquakes (M > 5.5). This hypothesis was corroborated by the observation that all the strongest shocks of the last three decades in the Italian territory are located in areas characterized by a high rate of deformation [17].

## 5. Conclusions

In the last twenty years, the main shallow earthquakes (depth $\leq$ 15 km) in Italy and the Alps have occurred only in some of the horizontal strain rate zones, as depicted by Montone and Mariucci [15]. Meanwhile, the strain rate is currently low in other areas affected by recent earthquakes that occurred before the 1990–2012 survey, such as Belice (1968) and Friuli (1976). These areas are also where the higher seismic events from 1915 to now have occurred. The area of the 1915 Marsica earthquake also shows lower-than-surroundings strain rate values, such as in the Central and Eastern sections of the Northern Apennines (in the Western sector, higher seismicity barely corresponds to a slightly higher strain rate). In this picture, the high strain rate level indicates that the scenario of a new strong shake in Irpinia is not unlikely. Additionally, the historical record is in agreement with this, given the short temporal distance between strong (M6+) seismic events in Irpinia during the 400 years of catalog completeness (i.e., from 1620 to present), and a long 200-years period without M6+ seismic events occurring between 1732 and 1930.

Moreover, by merging historical seismicity and InSAR satellite data, we think that, in the future, a hypothesis related to the following scenarios should be explored:

- the short time gap between strong events in Irpinia during the 1694–1732 and 1930–1980 periods is linked to periods of continuous high strain rates;
- instead, the long seismic gap (a lack of strong seismicity) between 1732 and 1930 could have originated from a strain rate drop after the 1732 earthquake.

**Author Contributions:** Conceptualization, A.P., F.B.; historical and geographical investigation, F.B., A.P.; methodology, software, validation, formal analysis, data analysis, G.F.; investigation, resources, writing—original draft preparation, writing—review and editing, visualization, F.B., A.P. and G.F.; supervision, project administration, A.P.; funding acquisition, F.B. All authors have read and agreed to the published version of the manuscript.

**Funding:** This research received no external funding.

**Data Availability Statement:** The Parametric Catalogue of Italian Earthquake CPTI15 (version 3.0) is free and available at: https://emidius.mi.ingv.it/CPTI15-DBMI15/description_CPTI15_en.htm (accessed on 30 January 2021). As to any other dataset or maps here provided in this study, they can be available upon request.

**Conflicts of Interest:** The authors declare no conflict of interest.

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
