# Peer review of "Assessing Current Seismic Hazards in Irpinia Forty Years after the 1980 Earthquake: Merging Historical Seismicity and Satellite Data about Recent Ground Movements"

_geosciences, doi:10.3390/geosciences11040168_

Round 1

Reviewer 1 Report

I have completed my review of the manuscript "Assessing Current Seismic Hazard in Irpinia Forty Years after The 1980 Earthquake Merging Historical Seismicity and Satellite Data about Recent Ground Movements".

I have some concerns on publishing it without a large rearrangement of the manuscriprt.

1) the manuscript do not treat seismic hazard, so please adjust the title.

2) Introduction is lacking. Please add a secrion to state what is the problem you are going to treat.

3) Many missing references to study on active faults and geodetic studies. The latters are often focused on the comparison of  geodetic strain rates and earthquake release, that is what actually is in this manuscript. Please, consider the D'Agostino's work and reference therein.

4) Conclusion are not supported by the results and their discussion. Then, even if you state them as "hypothesis", they have to be supported by a clear interpretation of the data. Please, you should treat these hypothesis in the discussion of the results before stating in the conclusions.

5) please add some paragraphs to explain how  strain rate is computed. There is only a reference to a previous work, but a few words here could help the reader.

Reviewer 2 Report

The research is valuable and the manuscript is interestingly written. The research itself makes an important contribution to the seismic hazard assessment. I am wondering to see the results of the method suggested by authors on bigger (spatially) dataset.

I have just several comments:

line 25: It is stated that a period of observation covers period between 1991 and 2011, but in Figs. 2, 3 and 6 and somewhere in the text it is mention 1990-2017 period?!

The main faults should be marked in Figure 2, especially those mentioned later in the text.

It would be very interesting to know the hypocentral depths of EQs, especially those occured since 1900.

line 86: The mentioned polygon should be marked in Figs. 2 and 3.

line 91: "for the strongest events" - of which magnitude (important!)?

Figure 4. – Translate the terms on the abscissa and ordinate into English, as well as the first sentence in the figure caption.

line 111: It is not quite obvious that the strongest EQs seem to tend to group together over time!!

Table 1. - After 1982 there were no EQs with M5+??

line 164: "right and left" should be "top and bottom" (or better "up and down").

line 184: "In the ..."

Something is missing in conclusion - for instance, it would be interesting to compare in this research obtained results with existing seismic hazard maps of the area od interest.

Reviewer 3 Report

Review of:

Assessing Current Seismic Hazard in Irpinia Forty Years After The 1980 Earthquake Merging Historical Seismicity And Satellite Data About Recent Ground Movements

By Aldo Piombino, Filippo Bernardini, Gregorio Farolfi

This paper attempts to assess the present day seismic hazard in Irpinia by combining seismological data from the 1980 earthquake and ground deformation from recent satellite data. The method is based on the analysis of a detailed ground velocity map of Italy determined by GNSS and InSAR interferometry data between 1991 and 2011.

After the presentation of published catalogue seismological data, and the “qualitative” comparison with also published geodesy data the authors conclude that the short time gap between strong events in Irpinia in the periods 1694 – 1732 and 1910– 1980 are linked to periods of continuous high strain rate and that the long seismic gap between 1732 and 1910 could have been originated by a strain rate drop after the 1732 earthquake.

The concept of merging historical with modern data is challenging nowadays and useful for the seismic hazard assessment of an area given the short instrumental era. This aspect is an advantage of this study.

However, the comparison between earthquakes and strain is only visual, without taking into account stress field and energy release from earthquakes, which could add a lot to the reliability of the suggested outcome. In other words there is not a quantitative linkage between the two datasets. Moreover, the two datasets are widely used in recent publications and well explained. Neither new data are offered by this work to the scientific community, nor novel knowledge on the study area.

Given the above, I suggest that the paper is not worth to be published in Geosciences, except the authors can clearly present the novelty and reliability of their work, if any.

Lastly, the following comments are given for the perusal of the authors.

L41: “emplacement of melt intrusions along lithospheric faults in the crust” this is wrong, there are no faults

L47: Change to “…presents major earthquakes from 1990 to the present…”

L48-49: Please modify this sentence, it is not clear. Perhaps you could delete it since you mention this effect previously.

L50: The main faults in Irpinia as well… Delete “as well”

L50: “belong”

L56-57: “…this channel shows prevalent west-directed velocities in the Stable Europe frame, nested in the Adriatic eastward moving block.” What do you mean with this sentence? How west-directed velocities are nested in the Adriatic eastward moving block? Please clarify.

L59-60: Why some of the earthquakes are tagged, while others not? Please explain better what Fig. 2 shows.

L68: Please depict “Irpinia dome” on a map

L69-71: The geographical axis of the Irpinian dome approximately corresponds to the isoline of 1 mm/a eastward in the E-W component of the motion. This sentence is not clear.

L71-73: “Any useful information of the N-S component of the ground movement can be detect by InSAR satellites because their quasi-polar orbits make possible only the detection of vertical and E-W velocity components.” How can you obtain information on the N-S component when only the detection of vertical and E-W velocity components is possible?

L78-80: Geographical description of the study area has been given previously. Please marge this information and place it in the introductory section.

L84-85: You probably mean “on the surface projection of the hanging-wall”

L105: “Correggere la legenda Year e Magnitude.” Please translate…

Fig. 4: Please explain in the caption what the different colors represent. Long spans of time lack information or earthquakes? This is arbitrary, can you provide historical evidence why lack of information?

L139-141: Linkage of high hazard with high strain rates is not a new theory. It is a reasonable and well known aspect. Please rephrase and give the novelty of your effort instead, this is something that lacks throughout the paper.

L141-147: How seismicity from other remote areas, affect Irpinia?

L145-147: Show toponymes on a map

L169-170: “above” the hanging wall of the Monte Marzano fault system

L173: “In the picture of the EW-trending lithospheric faults…” what do you mean by lithospheric faults, please define. Do you mean faults that cut the crust or faults that enter the mantle? The latter in terms of rock rheology is not possible.

Figure 7. One of the two panels should be removed. Toponymes fonts are illegible.

L191-192: “Instead in Irpinia, as seen, high strain rate has been detected, implying from a theoretical point of view a high possibility of new strong earthquakes in the area.” This is self-evident, cannot be a conclusion of the study.

L202-203: “the short time gap between strong events in Irpinia in the periods 1694 – 1732 and 1910– 1980 are linked to periods of continuous high strain rate;” How do you come to such a conclusion? You better use “could be linked” instead of “are linked”. This comment holds for similar cases throughout the document.

Round 2

Reviewer 1 Report

Abstract. I suggest to delete lines 11-13 and start from “Recently a new….”

L33. I suggest to name the section “introduction”.

L42. What is C2 ?

L76. There are no seismic sources, just epicentres

L85. Is Irpinia Dome composed by the two the cyan areas?  I see it is in the caption, however I sugegst to write also here.

-Why figure 2 bottom is not described before figure 3?

L94. What are PS? This acronym is not discussed in the text.

-Legend figure 3. You have a classified type legend, but actually the map uses graduation, please adjust.

L150. The new theory link the occurrence of earthquakes with high strain rate, not the seismic hazard. Seismic hazard is the hazard associated with potential earthquakes in a particular area. The theory do not gives information about magnitude and depth of future earthquakes, so how can you assess this hazard?

Figure 7 is not cited in the text. Why you map epicentres over a seismic hazard map of PGA with a probability of exceedance of 10 % in 50 years? You are mapping two different objects.

Figure 7. Please, adjust the legend.

L300. I’m not sure the current building code still uses the four classes.

L306. How can you be sure that the strain rate is decreased since 1968 in the Belice?

Author Response

Reviewer 1

Comments and Suggestions for Authors

Abstract. I suggest to delete lines 11-13 and start from “Recently a new….”

Answer: Thanks, we modified following the suggestion.

L33. I suggest to name the section “introduction”.

Answer: Thanks, we modified following the suggestion.

L42. What is C2 ?

Answer: C2  is a mathematical term that we explained in the manuscript.  C2 means a continuous bi-cubic interpolation function

L76. There are no seismic sources, just epicentres

Answer: Thanks, although we believe that the meaning of the sentence is clear (i.e. each epicenter is linked to a seismogenic source) we modified the phrase by using the term seismogenic areas.

L85. Is Irpinia Dome composed by the two the cyan areas? I see it is in the caption, however I

sugegst to write also here.
Answer: Yes, thanks. We modified the text as follow: The uplifting area is divided in two different parts, separated by a narrow corridor of lower uplift < 1 mm/a, (Fig. 3). it is interesting to note that this corridor is placed near the epicentral area of the 1930 and 1980 earthquakes.

-Why figure 2 bottom is not described before figure 3?

Answer: Thanks, we modified the figures and the text. Fig. 2 now consists of two images with separately toponymy and seismic events. Also fig. 3 consists of two images, one showing the East-West component and the other showing the vertical component. They are all described in the text.

L94. What are PS? This acronym is not discussed in the text.
Answer: Thanks, we modified the text following the suggestion.

-Legend figure 3. You have a classified type legend, but actually the map uses graduation, please adjust.

Answer: we used this simple and clear method commonly used to present the results. Fuzzy colours represent values in between the reference color of the legend.

L150. The new theory link the occurrence of earthquakes with high strain rate, not the seismic

hazard. Seismic hazard is the hazard associated with potential earthquakes in a particular area. The theory do not gives information about magnitude and depth of future earthquakes, so how can you assess this hazard?
Answer: Thanks, we modified the text as follow:  This new theory based on observables identifies significant earthquake (M>5.5) prone areas with high strain rate areas

Figure 7 is not cited in the text. Why you map epicentres over a seismic hazard map of PGA with a probability of exceedance of 10 % in 50 years? You are mapping two different objects.

Answer: We do not discuss this figure in the text. It is only meant to show that the current map of Italy's seismic hazard mainly derives from the historical seismic record. Epicenters of the main earthquakes overlap exactly with the areas of highest hazard.

Figure 7. Please, adjust the legend.
Answer: Thanks, we modified the legend.

L300. I’m not sure the current building code still uses the four classes.

Answer: Thanks, we modified the phrase.

L306. How can you be sure that the strain rate is decreased since 1968 in the Belice?

Answer: we stated that the current strain rate is low. There is no comparison with the historical strain rate of previous surveys.

Submission Date

26 October 2020

Date of this review

18 Dec 2020 08:59:39

Reviewer 3 Report

I have rejected this paper in the first round. In the revised version the authors have addressed most of the moderate and minor concerns. However, my main objection about the novelty of the method remains. I suggest that the authors undermine this issue and stress upon the significance of linking historical data with recent observations, to assess possible repetitions of damaging earthquakes. This is an interesting topic with a plethora of applications globally.

For the above reason, but having also noticed problems in the structure and content of the manuscript, I recommend resubmission and re-evaluation of the paper from scratch.

Author Response

reviewer 3

(x) I would not like to sign my review report

( ) I would like to sign my review report

English language and style

( ) Extensive editing of English language and style required

(x) Moderate English changes required

( ) English language and style are fine/minor spell check required

( ) I don't feel qualified to judge about the English language and style

Yes Can be improved Must be improved Not applicable

( ) (x) ( ) ( )

( ) (x) ( ) ( )

( ) (x) ( ) ( )

( ) (x) ( ) ( )

( ) (x) ( ) ( )

Comments and Suggestions for Authors

I have rejected this paper in the first round. In the revised version the authors have addressed most of the moderate and minor concerns. However, my main objection about the novelty of the method remains. I suggest that the authors undermine this issue and stress upon the significance of linking historical data with recent observations, to assess possible repetitions of damaging earthquakes.

This is an interesting topic with a plethora of applications globally.

For the above reason, but having also noticed problems in the structure and content of the

manuscript, I recommend resubmission and re-evaluation of the paper from scratch.

Answer to the Reviewer:
Our paper presents two novelties.

  1. First of all, we have determined the strain rate by the integration of GNSS with InSAR techniques. Results provide the velocity and displacement of millions of ground points identified by the Persistent Scatterers (PS). This method for the determination of the strain rate is absolutely new and the adoption and integration of GNSS and InSAR techniques provide the velocity and displacement of millions of ground points identified by the Persistent Scatterers (PS). Thus, such method implies highly finer restitution than the ones performed with only a few hundreds of GPS stations and, more, it allows observations also where/when there are no GPS station close to an area

It has been presented in 2020 on Scientific Reports: Farolfi, G., Keir, D., Corti, G. et al. Spatial forecasting of seismicity provided from Earth observation by space satellite technology. Sci Rep10,9696 (2020). https://doi.org/10.1038/s41598-020-66478-9
Here we present the very first application to a local situation of that research.

  1. The second novelty is the analysis of the linking between historical data and strain rate determined by such satellite observations. This is an original approach never done before.

Submission Date

26 October 2020

Date of this review

07 Dec 2020 12:09:49

1) the references about the 1980 earthquake must be expanded with this paper:

Paper suggested

Our answer

Papanikolaou and Roberts 2007

It has been interesting to read also other works of these authors. We have cited a work of the same authors published in 2011.

Ascione et al 2020

This work is interesting, but since it talks about structural geology and CO2 triggering, we think that it is outside the purposes of our work

Matano et al 2020

the same answer as for Ascione

Gizzi and Potenza

In the issues presented in this work (in figure 6) there are not “historical seismicity” and “strain rate”.  For us, it testifies the novelty of our work

Galli 2020

It has been useful, assigning both the 62 and 989 earthquakes to the Monte Marzano Fault

2) About the 1930 earthquake for a different idea of the fault we suggest to read also the paper of : 
Serva, L.; Esposito, E.; Guerrieri, L.; Porfido, S.; Vittori, E.; Comerci, V. Environmental effects from five historical earthquakes in southern Apennines (Italy) and macroseismic intensity assessment: Contribution to INQUA EEE Scale Project. Quat. Int. 2007, 173?174, 30-44. 

Answer: This work and other cited demonstrate that also for the 1930 earthquake the seismogenic source is in Apenninic direction. We have modified the text. Thank you

3) Figures 2, 3, 5, 6, 7: it is necessary to modify the labels because they overlap the

seismicity and other information contained in the figures.

Answer: Excuse us, but we had already modified some of these figures according to the suggestions you gave us in the first revision, and we do not understand why further modifications are now required. The superimposition of toponyms on earthquakes is inevitable, and occurs whenever one wants to represent the toponymy and distribution of earthquakes in the same figure.

That said, we have nevertheless tried to follow your suggestion and to make the figures more readable: Fig. 2 now consists of two images with separately toponymy and seismic events. Also fig. 3 consists of two images, one showing the East-West component and the other showing the vertical component. We then changed the text following the new numbering.

As for figures 6 (bottom) and 7 we have decided to eliminate some labels concerning the toponyms of minor places, leaving only the names of a few of the main cities in the studied area. Therefore, now these images should be more readable.

In Figure 6, bottom, and Figure 7 is necessary to replace Conza with Conza della Campania.

Answer: as mentioned here above, in these images we have decided to eliminate some labels concerning the toponyms of minor places (including Conza), leaving only the names of a few main cities in the studied area. Therefore, now these images should be more readable.

Moreover, in Figure 7 replace Volcei with Buccino that is the current name of the town.

Answer: as mentioned here above, in this figure we have decided to eliminate some labels concerning the toponyms of minor places (inlcuding Volcei), leaving only the names of a few main cities in the studied area. Therefore, now this image should be more readable

4) In Table 1 is necessary to add Io (MCS)

Answer: Done, thanks

5) We also suggest to discuss better the results, describing the quaternary faults (e. g. Val

D'Agri, Vallo di Diano, Cervialto) that are well documented (e.g. Roberts and

Papaninkolaou; Sgambato et al., and so on). Moreover the authors must say that Irpinia

Fault is one of the possible areas in that sector that are ready to make a destructive

earthquake. The authors could draw a parallel with Central Italy, say that now everyone is

worried about Sulmona, but Irpinia and the other faults in the southern Apennines, from the

geodetic and historical seismic point of view, show similar evidence.

Answer: This work deals with Irpinia only. Moreover, it’s beyond our scope the description of faults outside Irpinia. And we have not written something about the Vallo di Diano, but the same is for Sannio and Matese.

However, thanks to your suggestions, we have concluded the article including the illustration of the earthquake clustering in Northern and Central Apennines. The clustering of Apenninic earthquakes is presented in many works. Someone talks about “stress transfer, others of a “domino effect” among various blocks (eg. Mantovani et al).

We think that strain rate is the most powerful explanation for the clustering, for the motivation expressed in our text between figure 7 and the conclusions.

Actually, we are not strictly worried about Sulmona, even if we are concerned for the Venafro – Meta mountains (a NS 40 km-long area starting 20 km SSE from Sulmona:  Cassino, Venafro, Isernia and Pescasseroli are closer than Sulmona to this area) and for Central Northern Sicily.

It is also evident that  the epicentral areas of the 2009 – 2016 Central Apennines earthquakes in the 1990-2011 period were showing the highest strain rate.

Round 3

Reviewer 1 Report

I think  the manuscript can be considered ready for being published.

Author Response

thank you

Reviewer 3 Report

I would like to ask the authors the following, before the paper is published:

  1. Language check
  2. Undermine the fact that the method is novel, only the concept is tentatively novel.
  3. Point out some general remarks on what do we expect with regard to the structural damage in case of a future repetition of the 1980 earthquake

Author Response

The third reviewer finally agrees about the publication of this work, but he proposes other issues. We think that he don't want the article to be published for something he/she only knows.

1. language check.

We don't understand why now there are language problems that the revisor has not noted in the elder version of the work

2. Undermine the fact that the method is novel, only the concept is tentatively novel.

We completely disagree.

  1. Firstly, for us we have presented TWO novelties. Not a "method" and a "concept".

  2. Well, we are not the first Authors talking about strain rate (and we have described older works about this issue with other methods), but this method is new, and – more – it is the first application of a work that has demonstrated a link between strain rate and Magnitude, a relation until now impossible to verify with the GPS stations or fault slips only because of the amount of data.

  3. The comparison between strain rate and historical seismicity is also a new method, and we were able to demonstrate that the historical point of view and that of the strain rate both agree on the possibility of a new strong earthquake in Irpinia"

3. Point out some general remarks on what do we expect with regard to the structural damage in case of a future repetition of the 1980 earthquake:

We discuss only geological and physical constraint, not engineering issues. Yes, structural damage depends from many geophysical factors (e.g. hypocentral location, stratigraphic and structural situation, the possible polarization of seismic waves; but the main problem lies in how (and where!) manufacts are build. These considerations must be done by a structural engineering work. We only say to engineers: be careful about building in Irpinia because the high strain rate is a preferential situation for the occurrence of a strong earthquake

Moreover, the damage may also be partly dependent on the weather conditions of the previous weeks: because of the morphology and the soils of the region, in which landslides are the most important driver of the landscape changes, during a wet period the damage can be exacerbated by the occurrence of much more landslides than in a dry season. The damage increase because of the heavy rains of the weeks

The greater damage in landslides-prone areas due to the intense precipitations in the preceding weeks is clearly visible, for example, in the M 6.6 2018-09-05 Hokkaido and M 7.5 2018-09-28 Palu earthquakes